# The effects of language and emotionality of stimuli on vocabulary learning

Candice Frances[1,2]*, Angela de Bruin[3], Jon Andoni Duñabeitia[4,5]

**1** Basque Center on Brain, Language and Cognition, Donostia, Spain, **2** Department of Social Sciences and Law, UPV/EHU, Donostia, Spain, **3** Department of Psychology, University of York, York, United Kingdom, **4** Centro de Ciencia Cognitiva–C3, Universidad Nebrija, Madrid, Spain, **5** Department of Language and Culture, The Arctic University of Norway, Tromsø, Norway

* candice.frances@ncf.edu, c.frances@bcbl.eu

**Data Availability Statement:** All data used for this manuscript are available at https://osf.io/64b3e/?view_only=eac93c2fc41b42fb9cd3519466cd62af.

**Funding:** This research has been partially funded by the following entities: Spanish Government [https://ciencia.sede.gob.es/]: PGC2018-097145-B-

## Abstract

Learning new content and vocabulary in a foreign language can be particularly difficult. Yet, there are educational programs that require people to study in a language they are not native speakers of. For this reason, it is important to understand how these learning processes work and possibly differ from native language learning, as well as to develop strategies to ease this process. The current study takes advantage of emotionality—operationally defined as positive valence and high arousal—to improve memory. In two experiments, the present paper addresses whether participants have more difficulty learning the names of objects they have never seen before in their foreign language and whether embedding them in a positive semantic context can help make learning easier. With this in mind, we had participants (with a minimum of a B2 level of English) in two experiments (43 participants in Experiment 1 and 54 in Experiment 2) read descriptions of made-up objects—either positive or neutral and either in their native or a foreign language. The effects of language varied with the difficulty of the task and measure used. In both cases, learning the words in a positive context improved learning. Importantly, the effect of emotionality was not modulated by language, suggesting that the effects of emotionality are independent of language and could potentially be a useful tool for improving foreign language vocabulary learning.

## Introduction

Our current study addresses the question of whether emotionality—specifically, positive valence and high arousal—affects word learning in a native and a foreign language. In particular, the focus is on cases in which one's only experience with the to-be-learned object is a definition without having seen or touched that object in person. Although this question may seem odd, in reality, when we learn new information in academic settings, our experience with the content we learn is quite limited. If we take the example of biology or history class, the information we learn there, regardless of how concrete, will hardly ever be a part of our sensory experience. According to Kousta et al. [1], this should lead to poorer representations and more difficulty in acquisition. In cases like Content and Language Integrated Learning (CLIL), new information is also conveyed in a foreign language, adding to the complexity of learning and

I00 (JAD), RED2018-102615-T (JAD), SEV-2015-0490, and BES-2016-077169 (CF) Comunidad de Madrid [https://www.comunidad.madrid/]: H2019/HUM-5705 (JAD) The funders had no role in the study design, data collection and analysis, decision to publish, or preparation of the manuscript.

**Competing interests:** The authors have declared that no competing interests exist.

remembering information that is abstracted from sensory experience. Given this increased difficulty in learning, it is important to find tools that can improve or facilitate this process. One possible tool is emotionality, as it has been suggested to improve learning in other cases [1,2], but it is not clear whether it also be useful in this kind of experience. In particular, would having only verbal and not sensory experience with the content make emotionality useful even for acquiring new concrete nouns? Kousta et al.'s [1] theory would suggest that emotionality, given its independence from concreteness, would indeed play a facilitatory role in that case.

In this study, we therefore examined whether vocabulary learning in a native and foreign language can be facilitated by emotionality. We use the terms 'native' and 'foreign' rather than 'first language' and 'second language' because we focused on bilinguals living in a bilingual society who speak multiple languages in the community but also acquire another language (English) at school. In many (monolingual and bilingual) societies it is now common for children to acquire foreign languages at school. In some cases, the foreign language is taught through content courses in that language. Currently, most of the research addressing this type of learning in a foreign language comes from classroom studies on CLIL using children. This literature focuses on language outcomes more so than on content learning and has found mostly positive results in this area [3–11]. The literature on adult foreign language medium learning is more limited but also focuses on benefits associated with language [12] and not content. The research on content learning is both more restricted and less clear-cut, with studies showing positive, negative, and null results [13]. Overall, studies tend to focus on the efficiency of courses in general, evaluating overall performance at the end of the term. Very few studies compare the immediate understanding and learning of new content in a foreign and a native language. Those who do have found either detrimental or null effects of foreign language use, depending on the task [14,15], particularly without foreign language support [16]. These results have been accounted for in the context of cognitive load theory, which suggests a working memory overload for individuals trying to learn content in a language they are not proficient in [16]. Importantly, contributing to this literature would improve our understanding of whether foreign and native language learning differ as well as providing insights into possible improvements on foreign language teaching methods.

As mentioned before, we will focus on emotionality as a potential way of improving word learning. In particular, emotionality provides greater richness to lexical items, aiding their processing [1] and increasing familiarity at the time of retrieval. This process is facilitated by an enhancement of attention during encoding, leading to more durable memory [17]. This idea has been supported by studies on language processing in participants' native language [17]. These studies have shown that emotionality facilitates learning in the native language—with emotional words having lower ages of acquisition—as well as aiding processing and memory [1]. But this phenomenon is still poorly understood, especially in the case of bilinguals. In addition, research so far has focused mostly on how we process or respond to known emotional words, but we do not have a clear idea of how learning in general and vocabulary learning in particular are affected by the valence of the new word or concept. Therefore, it is unclear whether we learn words that refer to new emotional concepts better or easier than those referring to new neutral concepts.

In the foreign language, the story is not as simple. This is in part because of a decrease in emotionality in the foreign language [18]. Focusing on memory for known words, the effects of this decreased emotionality are very inconsistent. Some studies have found enhanced memory for emotional as compared to neutral words in a foreign language [19–23] while others have not [24]. In addition, others have even found inconsistent [25] or inverse effects [26].

The literature on emotionality and word acquisition in a foreign language is quite scarce. Most studies have focused on acquiring new lexical forms for known concepts. For example,

Ferré, Ventura, Comesaña, and Fraga [25] studied new lexical form learning in a foreign language through word paring—associating a new item with the native language word. Results showed an improvement in later word recognition and translation for abstract words that were emotional (positive or negative). This shows that new labels for known concepts carry the same emotionality effects as the original terms, but in that context, emotionality played a role only in the acquisition of new forms for abstract words.

The question of whether emotionality affects word learning in a non-native language relates to the foreign language effect: the idea that people are less biased and more emotionally distant in their foreign language [27]. It is hypothesized that this effect may be a consequence of learning the foreign language in an emotionally attenuated academic environment [28]. Therefore, understanding how emotionality affects new word learning in a foreign language—i.e., new concepts with their associated form, not just learning new labels for known words or remembering known words—has implications for understanding the foreign language effect. To test this explanation, we have to manipulate the circumstances or context of the first exposure to a word or concept. In the current study, we do exactly that: we teach participants new concepts for which they do not have an equivalent native language word. This allows us to see what the effects of emotionality are as it is conveyed only by the definition of a word, without any prior exposure to the word or concept, or knowledge of the object.

Most studies so far have examined how emotionality affects the acquisition of new labels for known objects rather than learning of new words for new concepts. One notable exception is the study by Brase and Mani [29], which showed emotionality effects for new negative words in an emotional Stroop task and a sentence completion task. In the foreign language condition, this occurred only in the emotional context, whereas in the native language, they showed an effect in both emotional and neutral contexts. But, it is worth noting that their recall results are in contrast with prior studies, with no emotionality effect in the native language and a restricted effect in the foreign language—only for negative words in the emotional context.

Frances et al. [30] took a similar approach to the one in the present study, where participants learn new information—in their case new content, instead of new vocabulary—that was embedded in a positive or a neutral semantic context. This contrasts with prior studies that focused either on memory for known words or on the acquisition of new lexical items for known concepts. They found that positively valenced content was remembered better than neutral content and, importantly, that this effect occurred in the native and foreign languages equally. As we are using a similar strategy of teaching new information—in this case, new objects and their names—in the foreign or native language, we expect similar results. Namely, we expect enhanced memory for positive items and a similar improvement in both languages.

## Current study

The foreign language literature has shown inconsistent effects of emotionality and is unclear about whether emotionality helps increase vocabulary in that language. Therefore, the purpose of the current study is to understand new concept acquisition, removing the influence of prior experience with that concept or item. Importantly, in the current study, new information is taught implicitly through descriptive texts, which are more naturalistic than single, isolated words. This provides a more accurate understanding of what happens in more realistic learning situations.

Other studies have manipulated the emotionality of the context through association with emotional images [31] or paralinguistic information [29]. For the purposes of the present work, we have operationally defined emotionality as positive valence and high arousal

manipulated through the semantic context of the new word. Therefore, we manipulated emotionality semantically by conveying descriptions of objects in either a positive or neutral manner.

Given that one of the possible applications for this research is classroom contexts with students of different ages, only positive and neutral descriptions were used, as negative materials would not be advisable for educational settings. Furthermore, some studies have found that negative stimuli do not produce the same effects as positive stimuli and can hinder performance [19,21], particularly for the surrounding neutral words as they capture attention, unlike positive words which aid performance in these cases [32–34].

In short, this study aims to understand the effects of positive valence—manipulated semantically—on vocabulary learning. One of the priorities in the current study is to use naturalistic classroom-type stimuli than most prior research. The experiments here use novel words and concepts conveyed through paragraph-long definitions. We would like to emphasize that although the results of this study may have practical applications, our goal is to understand how learning can be influenced from a theoretical point of view. The ultimate goal is to understand how we process and incorporate this type of information in a foreign language, in particular as it compares to our native language.

## Experiment 1

### Methods

**Participants.** Participants were recruited from the Participa platform at the Basque Center on Cognition, Brain and Language. This database includes extensive demographic information, information on language history and use, and any physical or cognitive impairments. All eligible participants received an email inviting them to participate. Participants were typical young adults from the Basque Country.

Fifty-six participants were tested, three participants were removed for low accuracy ($< 50\%$) in the old/new recognition task (two from the English condition and one from the Spanish condition) and 10 were removed for low accuracy ($< 33\%$) in the name matching task (five from the English condition and five from the Spanish condition). The remaining participants were 43 native Spanish speakers (24.14 years old, SD = 3.99) with an intermediate to high level of English (BEST—Basque, English, and Spanish Test—Interview scores of four or five out of five [35]).

Language was a between-subjects factor. Participants did only one language condition: the native language condition (i.e., Spanish: 23) or the foreign language condition (i.e., English: 20 participants). Groups were matched for age (see Table 1), gender (13 females in the Spanish

**Table 1. Variables matched between groups in Experiment 1 and values for Experiment 2.**

| | | | English Assessment | | LexTALE | | IQ | | Corsi Task | |
|---|---|---|---|---|---|---|---|---|---|---|
| | | Age | BEST | AOA | Spanish | English | Verbal | Non-verbal | Backward | Forward |
| Exp. 1 | English group | 24 (3) | 4.2 (0.4) | 6.0 (1.9) | 95% (5%) | 72% (10%) | 108 (9) | 113 (8) | 6.30 (1.56) | 6.50 (1.43) |
| | Spanish group | 25 (4) | 4.2 (0.4) | 6.4 (2.5) | 92% (7%) | 70% (11%) | 110 (7) | 112 (9) | 6.48 (1.65) | 6.70 (1.55) |
| | Total | 24 (4) | 4.2 (0.4) | 6.2 (2.2) | 93% (6%) | 71% (10%) | 109 (8) | 112 (9) | 6.40 (1.59) | 6.61 (1.48) |
| | T-value df | -.979 (41) | -.054 (41) | -.572 (41) | 1.54 (41) | .528 (41) | -.724 (41) | .307 (41) | -.363 (41) | -.427 (41) |
| | *p* value | .333 | .957 | .571 | .132 | .601 | .473 | .761 | .719 | .671 |
| Exp. 2 | Total | 26 (6) | 4.2 (0.4) | 6.4 (2.5) | 94% (6%) | 72% (10%) | 100 (30) | 105 (31) | 6.54 (1.59) | 6.72 (1.57) |

*Note*: Values reported are means with standard deviations in parentheses. AoA stands for age of acquisition. BEST refers to the scores on the English interview portion of the BEST [35].

**Table 2. Education level by group and experiment.**

| Educational Level | Language in Experiment 1 | | | Experiment 2 |
|---|---|---|---|---|
| | English | Spanish | Total | Total |
| High School | 6 | 8 | 14 | 2 |
| Postgraduate | 2 | 4 | 6 | 9 |
| Professional Training | 2 | 2 | 4 | 3 |
| University | 10 | 9 | 19 | 30 |
| Total | 20 | 23 | 43 | 54 |

*Note*: Figures represent the number of participants in each category.

group and 12 females in the English group), education level (see Table 2), student status (17 students in the Spanish group and 16 in the English group), verbal and nonverbal IQ (Kbit) [36], age of acquisition of English, and Spanish and English LexTALE [37,38] (see Table 2). In addition, all participants were trilingual, with the exception of two participants in the Spanish condition who were only bilingual.

The experiment was approved by the Ethical and Scientific committees at the Basque Center on Cognition, Brain and Language (BCBL) and all participants provided written consent to participate in the study (Approval number: 7209). They were compensated 8€ for their time.

**Stimuli.** The stimuli consisted of 46 images of invented objects that had been used in another study [39] and 92 disyllabic pseudowords (46 for names of objects, 46 for fillers used in testing phase—see S1 Table for the full list of pseudowords and their corresponding bigram frequencies by language). Object names were equally likely to be an English or Spanish word (e.g., chefio) as measured by average token bigram frequency for each word ($M_{SP}$ = 491, $SD_{SP}$ = 411; $M_{EN}$ = 594, $SD_{EN}$ = 374; $t(45)$ = 1.22, $p$ = .230, $BF_{01}$ = 3.14, $error \%$ = 8.24 x $10^{-6}$) using B-Pal for Spanish [40] and N-Watch for English [41].

The object names were presented in 46 descriptions of made-up objects, their origin, and their use (see Fig 1 for an example). A positive and a neutral version of each description was created. The details changed between versions, but there was always at least one characteristic in common between the two descriptions (e.g., "it is a type of maze"). Each description was translated to create an English and a Spanish version. Descriptions were matched on length by language ($M_{Spa}$ = 45.2, $SD_{Pos}$ = 3.21; $M_{Eng}$ = 45, $SD_{Neu}$ = 3.07; $t(45)$ = .349, $p$ = .728, $BF_{01}$ = 5.90, $error \%$ = 1.15 x $10^{-5}$) and emotionality ($M_{Pos}$ = 46, $SD_{Pos}$ = 3.82; $M_{Neu}$ = 44.8, $SD_{Neu}$ = 3.52; $t(45)$ = 1.62, $p$ = .111, $BF_{01}$ = 1.85, $error \%$ = 7.93 x $10^{-4}$). Paragraphs had a length of 39 to 54 words per description, $M$ = 45.7, $SD$ = 3.79. Length was also matched using a 2-way ANOVA with language and emotionality on length of paragraph ($p$'s > .05, see Table 3).

Each description was controlled for word frequency to match emotion conditions. Word frequency was computed by taking the average frequency of every adjective, noun, adverb, and verb (excluding auxiliary and modal verbs) in the paragraph. For Spanish, we used EsPal [42] and matched the two emotionality conditions ($t(45)$ = .507, $p$ = .614, $BF_{01}$ = 5.54, $error \%$ = 1.12 x $10^{-5}$, see Table 3 for means and standard deviations). For English, we used the frequency from the Hyperspace Analogue to Language [43] using the English Lexicon Project [44] and matched the two emotionality conditions ($t(45)$ = .525, $p$ = .602, $BF_{01}$ = 5.49, $error \%$ = 1.11 x $10^{-5}$, see Table 3).

Positive and neutral descriptions had significantly different valence (ANEW [45]) for both Spanish ($t(45)$ = 4.26, $p$ < .001, $BF_{01}$ = 218, $error \%$ = 2.41 x $10^{-5}$) and English ($t(45)$ = 7.86, $p$ < .001, $BF_{01}$ = 1.96 x $10^7$, $error \%$ = 8.24 x $10^{-13}$), see Table 3. Positive and neutral descriptions also had significantly different arousal (ANEW—Bradley & Lang [45]) for both Spanish ($t(45)$

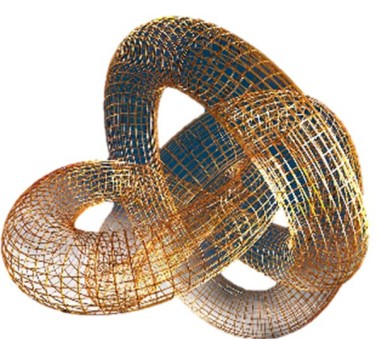

|  | | |
|---|---|---|
| Neutral | This is a chefio. The chefio is a type of maze that is used in research laboratories. The chefio has both closed and open parts so that people can search inside and orient outside. The chefio is a laboratory object. | Esto es un chefio. El chefio es un tipo de laberinto para laboratorios de investigación. El chefio tiene partes cerradas y abiertas para que la gente busque por dentro y se oriente al salir. El chefio es un objeto de laboratorio. |
| Positive | This is a chefio. The chefio is a type of maze used at birthday parties for children. The chefio has both closed and open parts providing the enjoyment and thrill of independence as well as safety of seeing their mother. The chefio is a game for children. | Esto es un chefio. El chefio es un tipo de laberinto para fiestas de cumpleaños infantiles. El chefio tiene partes cerradas y abiertas para dar el goce y la emoción de la independencia, así como la seguridad de ver a su madre. El chefio es un juego infantil. |

**Fig 1. Example stimulus.** All four versions of an example stimulus (neutral English, neutral Spanish, positive English, and positive Spanish) as well as the corresponding image. Due to copyright restrictions, the image is not one of the ones from our stimulus set, but it is similar to those we used.

= 5.01, $p < .001$, $BF_{01} = 2.02$ x $10^3$, $error \% = 2.46$ x $10^{-6}$) and English ($t(45) = 7.38$, $p < .001$, $BF_{01} = 4.19$ x $10^6$, $error \% = 1.85$ x $10^{-12}$), see Table 3. In addition, and in order to validate the emotionality (both valence and arousal) of the descriptions, we carried out a norming study with 18 participants from the same population as the study evaluating the descriptions in Spanish (see S1 Fig for instructions), on a scale from 0 to 4. The by item analysis found that items were considered significantly higher valenced in the positive version ($M_{Pos} = 3.27$, $SD_{Pos} = .335$) than the neutral version ($M_{Neu} = 2.21$, $SD_{Neu} = .352$), $t(45) = 15.48$, $p < .001$, $BF_{01} = 3.96$ x $10^{16}$, $error \% = 5.76$ x $10^{-21}$). Similarly, items were considered to cause significantly higher arousal in the positive version ($M_{Pos} = 1.32$, $SD_{Pos} = .454$) than the neutral version ($M_{Neu} = .459$, $SD_{Neu} = .394$), $t(45) = 8.56$, $p < .001$, $BF_{01} = 1.86$ x $10^8$, $error \% = 6.17$ x $10^{-11}$).

**Table 3. Means and standard deviations for control measures on the descriptions.**

|  |  | Frequency (words per million) | Words per description | Valence | Arousal |
|---|---|---|---|---|---|
| Spanish | Neutral | 153 (142) | 44.61 (3.69) | 5.86 (1.05) | 5.10 (.62) |
|  | Positive | 139 (108) | 45.80 (4.07) | 6.69 (1.11) | 5.71 (.71) |
| English | Neutral | 640 (258) | 44.89 (3.72) | 6.02 (.76) | 4.60 (.56) |
|  | Positive | 610 (334) | 45.72 (3.90) | 6.98 (.71) | 5.01 (.57) |

*Note*: Values in parentheses are standard deviations.

**Procedure.**   Participants were recruited from a database with some basic questionnaires: K-Bit—Kaufman Brief Intelligence Test, [36]—, LexTALE in English [37] and Spanish [38], linguistic background, socioeconomic background, and experience with bilingual education. These were used to select participants and match the groups. Then, they had a learning and a testing phase, which were either in English or Spanish (language was a between-subjects factor).

The learning phase consisted of seeing the picture of each object along with its description (containing the object name 4 times), one by one. Participants were instructed to read for comprehension. After seeing the object and reading the description for a minimum of 15 seconds (no Next button visible until the timeout) participants were allowed to move on to the next screen at their own pace. Participants took an average of 27.10 seconds (SD = 6.99 seconds) to read each description. The next screen asked them to type in the object's name. If they typed it incorrectly, they were taken back to the previous screen and then asked to type it again. This happened very infrequently (on average for two words for every three participants) and equally in every condition ($M_{Neu-Eng}$ = .186 words, $SD_{Neu-Eng}$ = .546; $M_{Pos-Eng}$ = .163 words, $SD_{Pos-Eng}$ = .374; $M_{Neu-Spa}$ = .163 words, $SD_{Neu-Spa}$ = .374; $M_{Pos-Spa}$ = .163 words, $SD_{Pos-Spa}$ = .374).

After seeing all 46 descriptions (23 positive and 23 neutral descriptions randomly mixed), they moved on to a filler task in order to reduce recency effects and assess working memory. The filler task consisted of a Corsi task (forwards and backwards) [46] which was timed to last 15 minutes. The purpose of the filler task was to decrease serial position effects [47,48] and reflect only information that had been more permanently registered in memory [49].

After the filler task, the testing phase started. The first task consisted of a cued recall in which participants were shown the image of an object and asked to recall its name (Fig 2A). This task was not timed. The second task was an old/new recognition task in which participants were presented the 46 object names along with 46 fillers one at a time and had 3000ms to say whether they had seen the word before or not (Fig 2B). The main task we were interested in, third task, consisted of a cued name-matching recognition task in which participants had 5000ms to select the correct name of the objects out of three choices (Fig 2C). For this task, the two distractors were pseudowords that had also been presented during the learning phase chosen randomly and matched for emotionality. This was our main task of interest because it was fully language independent—only the image and the name were presented, which were the same in both language conditions—and thus theoretically equally difficult. Finally, the last task was an attentional check that consisted of matching the correct characteristic that goes with the object (Fig 2D), with 5000ms to respond. The characteristic that had been held constant in both versions of the object description—positive and neutral—was used for this task. The two fillers were chosen at random from the other emotion-matched objects.

The order of tasks was chosen so that the earlier tasks would not affect the later tasks and held constant across participants. The name recall task was placed first because it was likely to be influenced by seeing the words again in the recognition tasks, but was unlikely to affect the other tasks. Then, in order to prevent increased familiarity with the learned items, we kept the Old/New task—which is purely based on familiarity—before the name matching task. Given that the attentional check was not of theoretical interest, we kept that task last.

## Results and discussion

The following are the linear mixed effect model analyses for both experiments. In all cases, analyses were run using linear mixed-effects models in R, lme4 [50] and lmerTest packages [51]. Both two-level categorical predictors (emotionality and language) were coded as -0.5 and

### (a) Task 1: Name Recall

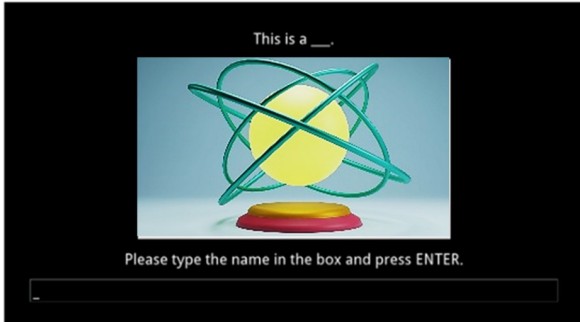

### (b) Task 2: Old/New Recognition

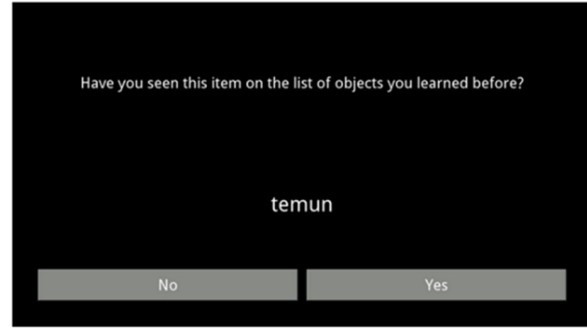

### (c) Task 3: Name Matching

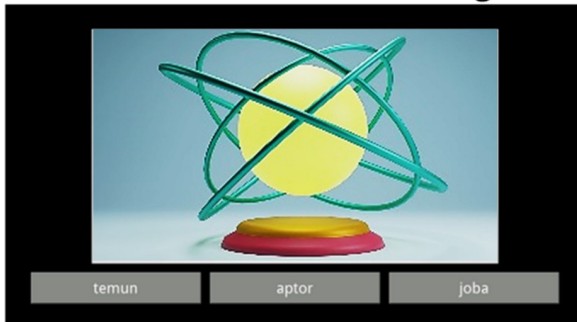

### (d) Task 4: Attentional Check

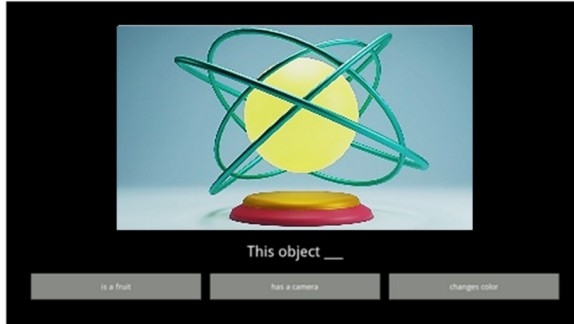

**Fig 2. Example screen for each of the four tasks in Experiment 1.** (a) Name recall task where participants were asked to type in the name of the object, (b) old/new recognition task where participants had to respond whether they had seen the word before or not, (c) name matching task where participants selected the correct name for the object, and (d) attentional check where participants selected the correct characteristic for the object. All responses were given using the keyboard (F and J keys for Old/New and A, F, and J keys for the matching tasks). Due to copyright restrictions, the image is not one of the ones from our stimulus set, but it is similar to those we used.

0.5 (neutral/positive and English/Spanish). Subjects and items were included as random effects, but due to convergence issues, random slopes varied by analysis. Models on response times have the log transformation of response time as the dependent variable.

The models included all fixed effects of interest, as well as random intercepts for participants and items, unless otherwise stated. When models did not converge, all correlations between the random slopes and the random intercepts were removed [52]. If the model still did not converge after removing the random correlations, we built down the random effects structure by removing the item slopes that explained the least variance until convergence was reached and singular fits were fixed.

Only the LME results are reported here. For the ANOVA results and the comparison between the two, please see (S1 Appendix).

**Name matching task.** The first model had accuracy (0 incorrect; 1 correct) as the dependent variable and language (English or Spanish) and emotionality (Neutral or Positive), as well as their interaction as fixed effects (see Table 4 for means, standard deviations, and 95% confidence intervals and Fig 3A for percentage of errors by condition). In addition, we included the random intercepts for subject and item. In the model, there was a significant main effect of emotionality ($\beta$ = .217, $SE$ = .092, z = 2.35, $p$ = .019), such that positive emotionality led to higher accuracy, and of language ($\beta$ = .245, $SE$ = .092, z = 2.65, $p$ = .008), such that Spanish led to higher accuracy. There was no interaction between emotionality and language ($\beta$ = .020, $SE$ = .183, $z$ = .108, $p$ = .914).

**Table 4. Means, standard deviations, and confidence intervals by participant for the name matching task by language and emotionality for Experiment 1.**

| | Accuracy | | Response Time | |
|---|---|---|---|---|
| | Neutral | Positive | Neutral | Positive |
| English | 0.42 (0.13) [0.36–0.47] | 0.47 (0.11) [0.42–0.51] | 2568.82 (669.92) [2275.21–2862.42] | 2394.26 (604.78) [2129.2–2659.31] |
| Spanish | 0.48 (0.11) [0.43–0.52] | 0.54 (0.1) [0.49–0.58] | 2514.84 (525.44) [2300.10–2729.58] | 2358.78 (539.34) [2138.36–2579.20] |

*Note*: Values in parentheses are standard deviations. The values in brackets are the 95% confidence intervals. Accuracy is in percent correct; response times are in milliseconds.

The model with log transformed response time as the dependent variable had language (English or Spanish) and emotionality (Neutral or Positive), as well as their interaction as fixed effects (see Table 4 for means, standard deviations, and 95% confidence intervals and Fig 3B for average response times by condition). In addition, we included the random intercepts for subject and item and random slopes for emotionality by subject. In the model, there was no main effect of emotionality ($\beta$ = -.042, $SE$ = .027, $t$ = -1.53, $p$ = .137) or of language ($\beta$ = -.005, $SE$ = .025, $t$ = -.215, $p$ = .830) and no interaction between emotionality and language ($\beta$ = -.023, $SE$ = .047, $t$ = -.496, $p$ = .620). (See S1.2 Table in S1 Appendix for the 90% confidence intervals of the effect sizes.)

Given the null interactions, we followed up the analyses using Bayesian statistics on the by-subject averages. Analyses were run using JASP statistical software (version 0.12.2) [53]. We used the default settings for all model comparisons (the priors were r scale fixed effects of .5, r scale random effects of 1, and r scale covariates of .354). For the t-test the prior was a Cauchy distribution of scale .707.

First, we analyzed the effects on accuracy. We ran a comparison between a model predicting accuracy from Language, Emotionality, and Subject to one which also included the interaction (Language x Emotionality). The model with the additional factor—the interaction—showed evidence that the data was approximately three times more likely to be observed under the null model without the interaction compared to the one with it, $BF_{01}$ = 3.33, *error %* = .485 (by item, $BF_{01}$ = 4.65, *error %* = 2.26), suggesting that accuracy was not affected by an interaction between language and emotionality. In a similar vein, we ran a t-test comparing the emotionality effect (accuracy on the positive condition minus accuracy on the neutral condition) and found moderate evidence that the emotionality effect was in fact the same between languages, $BF_{01}$ = 3.31, *error %* = .016 (by item, $BF_{01}$ = 6.08, *error %* = 1.67 x $10^{-5}$).

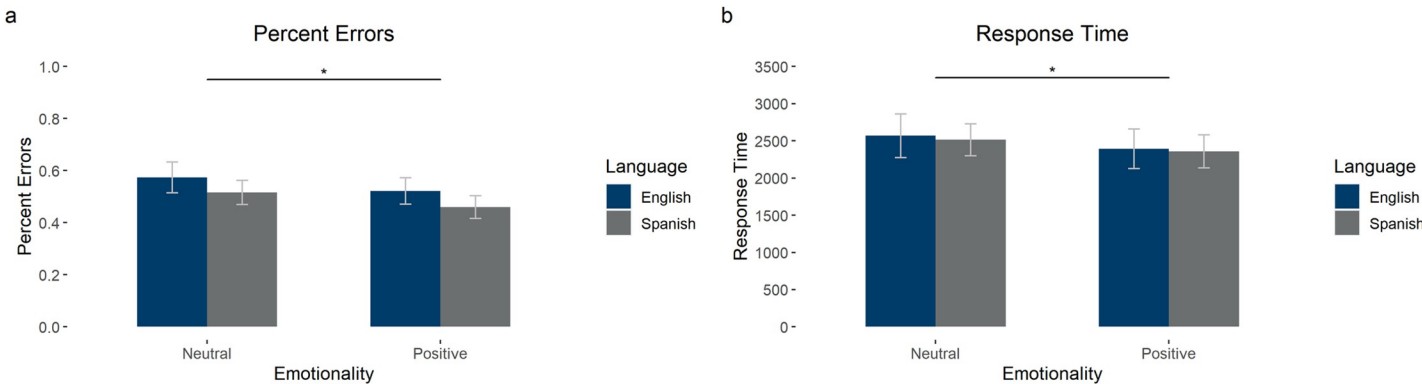

**Fig 3. Results from the name matching task.** The graphs show (a) percentage of errors and (b) response time. Error bars show 95% confidence intervals.

We then ran the same analyses on response time. We ran a comparison between a model predicting response time from Language, Emotionality, and Subject to one which also included the interaction (Language x Emotionality). The model with the additional factor—the interaction—did not show an improvement, $BF_{01}$ = 3.29, error % = .365 (by item, $BF_{01}$ = 2.80, error % = 4.39), suggesting that the interaction did not have an effect on response time and that the data was moderately more likely to be observed under the model without the interaction rather than the one with it. In a similar vein, we ran a t-test comparing the emotionality effect (response time on the positive condition minus response time on the neutral condition) and found moderate evidence that the emotionality effect was in fact the same between languages, $BF_{01}$ = 3.30, error % = .016 (by item, $BF_{01}$ = 4.43, error % = 1.09 x $10^{-5}$).

**Name recall task.**    The name recall task was evaluated in two ways: exact correct recalls and Levenshtein distance (LD) [54] between the produced string and the original item. For the first analysis, the number of exact recollections (correct word with the correct object) were counted for each participant within each emotionality condition. As expected, given the difficulty of the task, recall was very low [14], on average fewer than two words (M = 1.91 words, SD = 2.75 words).

The first model had accuracy (0 incorrect; 1 correct) as the dependent variable and language (English or Spanish) and emotionality (Neutral or Positive), as well as their interaction as fixed effects (see Table 5 for means, standard deviations, and 95% confidence intervals by condition). In addition, we included the random intercepts for subject and item. In the model, there was no main effect of emotionality ($\beta$ = .429, SE = .296, z = 1.45, p = .147) but there was one of language ($\beta$ = 1.21, SE = .306, z = 3.94, p < .001), such that participants recalled more words in Spanish than in English, and no interaction between emotionality and language ($\beta$ = .577, SE = .586, z = .985, p = .325).

Then, we calculated the normalized LD for each response. All responses that were shorter than 3 characters or contained a real word related to the object were removed as they were not considered real attempts. When calculating the LD, the number of insertions, deletions, and alterations needed to get from the produced word—the recalled string—to the original word—the studied string—is taken into account to calculate a standardized value. Then, these values are divided by the word length in order to normalize the value [55–57]. We also ran a model on the LD between the produced word and the correct response. The dependent variable was the normalized LD and language (English or Spanish) and emotionality (Neutral or Positive), as well as their interaction were the fixed effects. In addition, we included the random intercepts for subject and item. In the model, there was no main effect of emotionality ($\beta$ = -.179, SE = .131, t = -1.30, p = .195) but there was one of language ($\beta$ = -.374, SE = .142, t = -2.63, p = .008), such that participants produced strings closer to the correct one in Spanish than in English, and no interaction between emotionality and language ($\beta$ = .082, SE = .249, t = .330, p = .742).

**Old/new recognition task.**    The first model had accuracy (0 incorrect; 1 correct) as the dependent variable and language (English or Spanish) and emotionality (Neutral or Positive),

**Table 5.  Means, standard deviations, and confidence intervals by participant for the name recall task by language and emotionality for Experiment 1.**

|  | Number of words | | Normalized LD | |
|  | Neutral | Positive | Neutral | Positive |
|---|---|---|---|---|
| English | 0.45 (0.95) [0.04–0.86] | 0.45 (0.69) [0.15–0.75] | 0.76 (0.18) [0.68–0.83] | 0.77 (0.18) [0.69–0.84] |
| Spanish | 1.00 (1.48) [0.40–1.6] | 1.78 (2.24) [0.87–2.70] | 0.64 (0.23) [0.54–0.73] | 0.60 (0.27) [0.49–0.71] |

*Note*: Values are in number of words. Numbers in parentheses are standard deviations. The values in brackets are the 95% confidence intervals. Normalized LD values are in distance units with a range from 0 to 1, with 0 being the correct answer, identical to the target word, and 1 being a completely different word or no response.

**Table 6. Means, standard deviations, and confidence intervals by participant for the old/new task by language and emotionality for Experiment 1.**

| | D' | | Response Time | |
|---|---|---|---|---|
| | Neutral | Positive | Neutral | Positive |
| English | 0.78 (0.1) [0.74–0.83] | 0.8 (0.06) [0.78–0.83] | 1008.41 (189.65) [925.29–1091.52] | 961.81 (191.79) [877.75–1045.87] |
| Spanish | 0.81 (0.07) [0.78–0.84] | 0.8 (0.08) [0.77–0.84] | 1152.14 (392.57) [991.7–1312.57] | 1098.78 (401.61) [934.64–1262.91] |

*Note*: Values in parentheses are standard deviations. The values in brackets are the 95% confidence intervals. Response times are in milliseconds.

as well as their interaction as fixed effects (see Table 6 for means, standard deviations, and confidence intervals by condition). In addition, we included the random intercepts for subject and item. In the model, there was no main effect of emotionality ($\beta$ = -.085, $SE$ = .103, z = -.829, $p$ = .407) or of language ($\beta$ = -.024, $SE$ = .108, z = -.227, $p$ = .820) and no interaction between emotionality and language ($\beta$ = .115, $SE$ = .204, z = .561, $p$ = .575).

The model with log transformed response time as the dependent variable had language (English or Spanish) and emotionality (Neutral or Positive), as well as their interaction as fixed effects. In addition, we included the random intercept for subject but had to remove the random effects by item in order for the model to converge. In the model, there was a significant main effect of emotionality ($\beta$ = -.046, $SE$ = .018, $t$ = -2.55, $p$ = .011), such that positive emotionality led to faster response times, and of language ($\beta$ = .079, $SE$ = .020, $t$ = 3.96, $p < .001$), such that Spanish led to higher accuracy. There was no interaction between emotionality and language ($\beta$ = -.017, $SE$ = .036, $t$ = -.488, $p$ = .626).

The effects of language and emotionality did not appear consistently in all measures—they were present for name matching accuracy, but not response time; old/new response time, but not accuracy; and name recall only showed an effect of language. Importantly, we consistently found no interaction between language and emotionality, but need further evidence to verify that this interaction in fact does not exist. In addition, the language comparison was between subjects and the task was quite difficult in general. This suggests that the results need to be conceptually replicated. The next experiment addresses this and provides a close conceptual replication of this experiment.

## Experiment 2

The main goal of Experiment 2 was to replicate the results of Experiment 1 with a slightly different design. Besides that main goal, there were several minor goals. First, we wanted to have a fully within-subjects design in order to increase power, in case there was an interaction we were unable to detect before. Second, we wanted to improve the design and performance—in particular in the name recall task—by reducing the number of items to be learned per session. Although the total number of items was similar, participants learned only half in each of 2 sessions and the duration of the distractor task was reduced. Third, we wanted to remove the possible influence of the old/new task, which was not essential for answering our question. The attentional check also was not essential, but was maintained (at the end of the experiment) in order to (1) continue to assess whether participants had read the texts in their entirety and (2) to make sure that between session 1 and session 2 they were equally motivated to pay attention to the full texts, not just the object names.

### Methods

**Participants.** Sixty participants from the same pool as Experiment 1 were tested. One participant was excluded for not following directions, four more were removed for low accuracy

in the name matching task (<50% accuracy in either language), and one more was removed for a technical error. The remaining participants were 54 (17 male) native Spanish speakers (25.98 years old, SD = 6.19) with an intermediate to high level of English (as in experiment 1). Participants were not significantly different (*p*'s > .05) from those in Experiment 1 on any of the following variables: age, gender, education level, student status, verbal and nonverbal IQ (Kbit) [36], age of acquisition of English, and Spanish and English LexTALE [37,38]. As in Experiment 1, all participants were at least trilingual, with the exception of three participants that were only bilingual.

The experiment was approved by the Ethical and Scientific committees at the BCBL and all participants provided written consent to participate in the study (Approval number: 11709). They were also compensated for their time with 12€ after finishing the second session.

**Stimuli.** Stimuli were the same as in Experiment 1, except for six items (bangel, dela, fortor, mova, testor, and trequi). These items were removed in order to reduce the stimuli to 10 items per condition (neutral Spanish, positive Spanish, neutral English, and positive English).

**Procedure.** Participants were recruited from the same database as in Experiment 1. Each participant completed two sessions: one in English (foreign language) and one in Spanish (native language). The two sessions were at least 5 days apart and the order of sessions was counterbalanced between participants. The learning phase was the same as in Experiment 1. Participants took an average of 32.07 seconds (SD = 8.57 seconds). Again, participants had to retype the words very infrequently and equally in every condition ($M_{Neu-Eng}$ = .63, $SD_{Neu-Eng}$ = 1.12; $M_{Pos-Eng}$ = .52, $SD_{Pos-Eng}$ = .86; $M_{Neu-Spa}$ = .50, $SD_{Neu-Spa}$ = 1.13; $M_{Pos-Spa}$ = .24, $SD_{Pos-Spa}$ = .61). After seeing the 20 descriptions (10 positive and 10 neutral descriptions randomly mixed) for that session, they moved on to a filler task—forwards Corsi in session 1 and backwards Corsi in session 2 [46]—which was programmed to last 7 minutes.

After the filler task, the testing phase started and concerned only the 20 items from that session. The first task consisted of a cued recall identical to that of Experiment 1 (Fig 4A). The second task consisted of a cued name-matching recognition task similar to that of Experiment 1, except that participants had 2500ms to select the correct name of the objects out of two choices (Fig 4B). The response options were reduced to 2 in order to increase the reliability of response times. For this task, the distractors were pseudowords that had also been presented during the learning phase, chosen randomly. For each item there were 2 trials: one with a distractor of equal emotionality and one with the distractor of opposite emotionality.

The last task, the attentional check, was modified to display the object name rather than the image when asking for the correct characteristic (Fig 4C). The reason behind this change is that some of the characteristics were quite visual, making the discrimination deducible by the image and not dependent on the description. In any case, this task was kept only as a check. Participants had only two response options and 3500ms to respond. As with the name matching task, there were 2 trials for each item, one with a distractor of equal emotionality and one with the distractor of opposite emotionality.

## Results and discussion

All analyses were run as in Experiment 1. Only the LME results are reported here. For the ANOVA results and the comparison between the two, please see (S1 Appendix).

**Name matching task.** The first model had accuracy (0 incorrect; 1 correct) as the dependent variable and language (English or Spanish) and emotionality (Neutral or Positive), as well as their interaction as fixed effects (see Fig 5A and Table 7 for means, standard deviations, and 95% confidence intervals). In addition, we included the random intercepts for subject and item and the language slopes by participant (uncorrelated). In the model, there was a

(a) Task 1: Name Recall Task

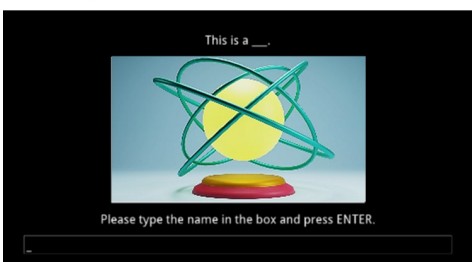

(b) Task 2: Name Matching Task

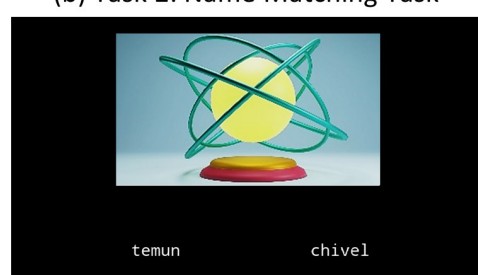

(c) Task 3: Attentional Check

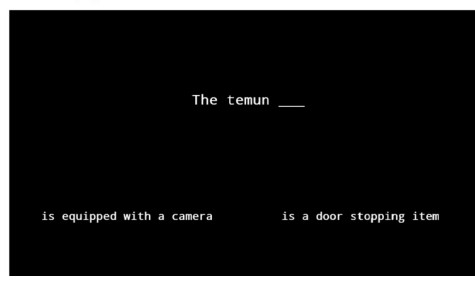

**Fig 4. Example screen of each of the three tasks in Experiment 2.** (a) Name recall task in which participants were asked to type-in the name of the object, (b) name matching task in which participants selected the correct name for the object, and (c) attentional check in which participants selected the correct characteristic for the object. In Tasks 2 and 3, participants responded using the keyboard keys F for left and J for right. Due to copyright restrictions, the image is not one of the ones from our stimuli, but it is similar to those we used.

significant main effect of emotionality ($\beta$ = .446, $SE$ = .150, $z$ = 2.98, $p$ = .003), such that positive emotionality led to higher accuracy, but no effect of language ($\beta$ = .170, $SE$ = .135, $z$ = 1.26, $p$ = .209) and no interaction between emotionality and language ($\beta$ = -.112, $SE$ = .153, $z$ = -.732, $p$ = .464).

The model with log transformed response time as the dependent variable had language (English or Spanish) and emotionality (Neutral or Positive), as well as their interaction as fixed effects (see Fig 5B and Table 7 for means, standard deviations, and 95% confidence intervals). In addition, we included the random intercepts for subject and item and random slopes for emotionality by subject. In the model, there was a main effect of emotionality ($\beta$ = -.065, $SE$ =

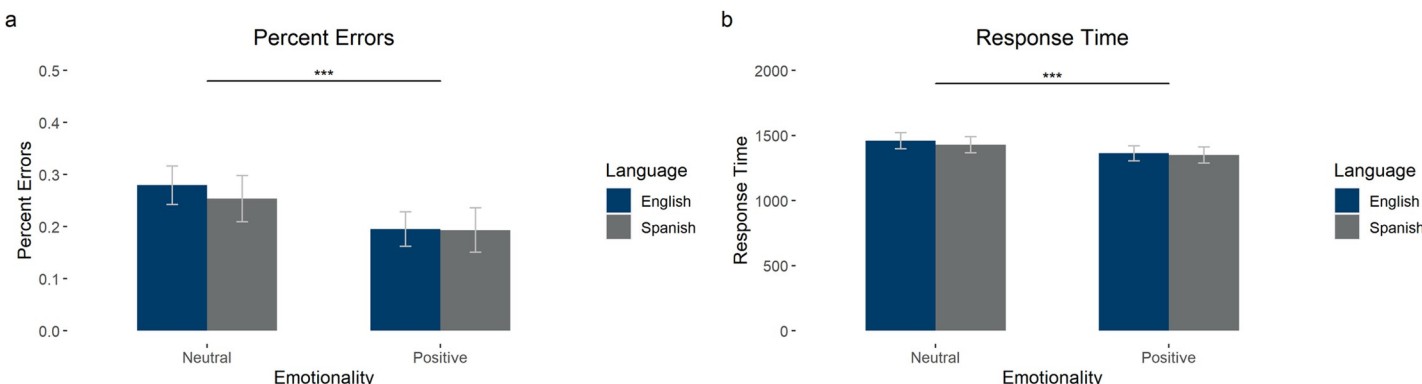

**Fig 5. Results from the name matching task.** The graphs show (a) percentage of errors and (b) response time in milliseconds. Error bars show 95% confidence intervals.

**Table 7. Means, standard deviations, and confidence intervals for the name matching task by language and emotionality for Experiment 2.**

| | Accuracy | | Response Time | |
|---|---|---|---|---|
| | Neutral | Positive | Neutral | Positive |
| English | 0.72 (0.14) [0.68–0.76] | 0.81 (0.12) [0.77–0.84] | 1605.37 (225.09) [1545.33–1665.41] | 1501.73 (219.57) [1443.16–1560.29] |
| Spanish | 0.75 (0.17) [0.70–0.79] | 0.81 (0.16) [0.76–0.85] | 1587.76 (250.04) [1521.07–1654.45] | 1499.36 (255.88) [1431.12–1567.61] |

*Note*: Values in parentheses are standard deviations. The values in brackets are the 95% confidence intervals. Accuracy is in percent correct; response times are in milliseconds.

.020, $t$ = -3.21, $p$ = .003), such that positive items were responded to faster, and of language ($\beta$ = -.026, $SE$ = .009, $t$ = -3.00, $p$ = .003), such that participants responded faster in Spanish than in English, but no interaction between emotionality and language ($\beta$ = .007, $SE$ = .017, $t$ = .403, $p$ = .687).

As in Experiment 1, we followed up the null interactions using Bayesian statistics on the by-subject averages using the same software and priors.

First, we analyzed the effects of the interaction on accuracy. We ran a comparison between a model predicting accuracy from Language, Emotionality, and Subject to one which also included the interaction (Language x Emotionality). The model with the additional factor—the interaction—showed evidence that the data was almost four times more likely to be observed under the model without the interaction compared to the one with it, $BF_{01}$ = 3.84, *error %* = .50 (by item, $BF_{01}$ = 2.13, *error %* = 1.22), suggesting that accuracy was not affected by an interaction between language and emotionality. We ran a t-test comparing the emotionality effect (accuracy on the positive condition minus accuracy on the neutral condition) and found moderate evidence that the emotionality effect was in fact the same between languages, $BF_{01}$ = 5.40, *error %* = .009.

We then ran the same analyses on response time. We ran a comparison between a model predicting response time from Language, Emotionality, and Subject to one which also included the interaction (Language x Emotionality). The model without the additional factor—the interaction—was almost four times more likely compared to the one with it, $BF_{01}$ = 3.81, *error %* = 3.75 (by item, $BF_{01}$ = 2.81, *error %* = .563), suggesting that the interaction did not have an effect on response time and that the data was moderately more likely to be observed under the model without the interaction rather than the one with it. We ran a t-test comparing the emotionality effect (response time on the positive condition minus response time on the neutral condition) and found moderate evidence that the emotionality effect was in fact the same between languages, $BF_{01}$ = 4.69, *error %* = .019.

**Name recall task.** The name recall task was evaluated as in Experiment 1. For the first analysis, the number of exact recollections (correct word with the correct object) were counted for each participant within each emotional condition. As expected, given the difficulty of the task, recall was very low [14]; on average fewer than four words out of the 40 ($M$ = 3.80 words, $SD$ = 3.80 words) were correctly recalled. The first model had accuracy (0 incorrect; 1 correct) as the dependent variable and language (English or Spanish) and emotionality (Neutral or Positive), as well as their interaction as fixed effects (see Table 8 for means, standard deviations, and confidence intervals). In addition, we included the random intercepts for subject and item. In the model, there was no main effect of emotionality ($\beta$ = .385, $SE$ = .281, $z$ = 1.37, $p$ = .170) nor of language ($\beta$ = .195, $SE$ = .153, $z$ = 1.27, $p$ = .203), and no interaction between emotionality and language ($\beta$ = -.107, $SE$ = .305, $z$ = -.349, $p$ = .727).

We also ran a model on the Levenshtein distance between the produced word and the correct response. The dependent variable was the normalized Levenshtein distance and language

**Table 8. Means, standard deviations, and confidence intervals for the name recall task by language and emotionality for Experiment 2.**

| | Number | | LD | |
|---|---|---|---|---|
| | Neutral | Positive | Neutral | Positive |
| English | 0.74 (1.03) [0.47–1.01] | 1.00 (1.26) [0.66–1.34] | 0.8 (0.14) [0.76–0.83] | 0.77 (0.15) [0.73–0.81] |
| Spanish | 0.93 (1.3) [0.58–1.28] | 1.13 (1.39) [0.76–1.5] | 0.76 (0.16) [0.72–0.8] | 0.72 (0.17) [0.68–0.77] |

*Note*: Values are in number of words. Numbers in parentheses are standard deviations. The values in brackets are the 95% confidence intervals. Normalized LD values are in distance units with a range from 0 to 1, with 0 being the correct answer, identical to the target word, and 1 being a completely different word or no response.

(English or Spanish) and emotionality (Neutral or Positive), as well as their interaction were the fixed effects. In addition, we included the random intercepts for subject and item. In the model, there was no main effect of emotionality ($\beta$ = -.030, $SE$ = .023, $t$ = -1.29, $p$ = .204) but there was one of language ($\beta$ = -.043, $SE$ = .013, $t$ = -3.23, $p$ = .001), such that participants produced strings closer to the correct one in Spanish than in English, and no interaction between emotionality and language ($\beta$ = -.009, $SE$ = .026, $t$ = -.338, $p$ = .735).

**Attentional check.** Finally, the average score on the attentional check for English was 73.2% (SD = 11.5%) and for Spanish it was 73.5% (SD = 11.0%). We used both a frequentist and a Bayesian paired samples t-test of language on accuracy by subject to verify that participants read equally well in both language conditions. The test showed moderate evidence that the null model was more likely than the alternative model, $t_1(53)$ = .22, $p$ = .829, Cohen's D = .029, $BF_{01}$ = 6.59, error % = .001, $t_2(38)$ = .64, $p$ = .525, Cohen's D = .203, $BF_{01}$ = 5.72, error % = 9.91 x $10^{-6}$.

**Discussion.** Overall, the results of Experiment 2 show a more consistent effect of emotionality in recognition, but not in recall. The effect of emotionality appears in the name matching task, both for accuracy and response time, but not in the name recall task. The effects of language are not very consistent, they appear in the name recall task—but, only for the Levenshtein distance measure—and for response time in the name matching task. Importantly, there was consistently no interaction between language and emotionality.

## General discussion

The motivation for this study was to shed light on a highly common phenomenon: learning new content in a foreign language. This occurs for example in CLIL-based school environments or in university programs taught in language other than the official one in the region. In particular, the focus was on using naturalistic materials and tests and to understand whether learning in a foreign language differs from learning in a native language. We focused on the effects of emotional context—here defined as positive valence and high arousal—during the process of learning new words (in this case, pseudowords) attached to new content, here represented by novel objects.

Importantly, all of the strings that were learned had no prior associations for the participants, as they corresponded to concepts invented for the purpose of this study, and they were equally likely in both languages. This means that any emotionality effects were caused directly by the semantic context created in the experiment, removing the possible impact of a priori developed language-emotion connections. We focused specifically on the acquisition of lexical forms and their association with an object image. Furthermore, these new words or strings were embedded in descriptions or definitions of the objects they represented, much like how we normally acquire new information.

We hypothesized that new lexical items would be more difficult to learn in a foreign language—even though the target stimuli were exactly the same between language conditions—

and that positive emotionality would help compensate for this difficulty. This was addressed through two experiments in which participants carried out a name matching task between the images of the objects they studied with their names (among other tasks). This task tested the associations that were created during the learning phase. We focused on this task because it was language independent, and thus exactly the same for both language conditions and not as taxing for participants. In addition, participants did a lexical decision task (old/new recognition task) in Experiment 1 and a name recall task in both experiments.

In Experiment 1, we found foreign language effects in the three main tasks: name matching accuracy, old/new response time, and name recall (both accuracy and Levenshtein distance). The critical task—the name matching task—showed a language effect on accuracy in Experiment 1, when there was more information to be remembered. This effect was present in response times—but not accuracy—in Experiment 2, when the task was made easier and the to-be-remembered information was reduced. These results align with prior studies that found improved performance in the native language when different pieces of information need to be associated [58], but the diverging findings depending on the tasks and measures used also provide a possible explanation as to why other studies do not find the same language effects [3,5,9,59]. Although there were several differences between the two versions of the task, most differences related to making the task less demanding, for example by reducing the number of items to be remembered, the retention period, the number of tasks, and the number of answer options. Given that the language effects remained in the Levenshtein measure in recall—with recall being the most demanding of the tasks—, one could interpret that the difficulty of the task overall affected accuracy disproportionately in the foreign language. It is worth noting that, although the two groups in Experiment 1 were very well matched, Experiment 2 was within subjects, which could have also influenced the results. Nevertheless, given that the only task in Experiment 2 that showed a language effect was the most difficult one—namely, recall —, this would support our idea that difficulty drove the effect. In order to conclude this with certainty, this hypothesis needs to be explored further in future studies.

With respect to emotionality, we found that positive semantic contexts aided recognition memory and, in particular, in the name matching task. These data fit well with preceding studies showing that learning performance is better when the information they are tested on is positive [18]. These effects of positive semantic context were present in both experiments, but absent in the recall tasks—possibly due to a lack of power in this task. In this case, unlike with the effect of foreign language, it was not modulated by the amount of information that had to be learned. This means that it is a more stable effect that is robust to some manipulations of other variables—such as language (as we discuss below) or amount of information. This suggests that this manipulation—positive valence and arousal of the semantic context—can be used in several different circumstances to increase familiarity with learned items. It should be noted though, that these results do not inform our understanding of the effects of negative valence on word learning and are not definitive with respect to recall.

When analyzing the size of the observed effects, the partial eta squared of the effects found in this study are considered to be medium to large effects [60]. Depending on the task and experiment, 7 to 46% of the variance was explained by emotionality (see S1 Appendix). For accuracy in the name matching task, this was 13% in Experiment 1 and 34% in Experiment 2 —with the latter more than doubling the 14% benchmark of a large effect. For response times in that task, effect sizes were 15% for Experiment 1 and 46% in Experiment 2, both considered large effects. In practical terms, emotionality lead to a 5 (Experiment 1) and 7% (Experiment 2) difference in accuracy, which equated to roughly remembering 2 to 3 items more. Although a relatively discrete boost in performance, this could be the difference between passing and failing an exam for a student that is struggling in a class and could become more meaningful as

it accumulates. With respect to response times, there was a 165 ms (Experiment 1) and 98 ms (Experiment 2) difference for this task. In more general terms, this study suggests that emotionally loaded semantic contexts—not just emotional content—conveying new pieces of information can improve short term recognition.

With respect to the relationship between emotionality and foreign language, we found no interaction in either accuracy or response times in the name matching task—or in any of the other tasks or measures—in either experiment. This means that, although learning new words and concepts may be more demanding in a foreign language—depending on the task—, the effects of emotionality remain constant across languages. Similar results have been found by other studies in the context of memory for known words [19–23], but the current study is the first to demonstrate this in the acquisition of new words and concepts as well as using more naturalistic stimuli.

These results may seem surprising, as the foreign language effect suggests that people are less emotional—and thus less affected by emotional biases—in a foreign language [61]. Nevertheless, our results do not contradict this idea, but rather expand on possible causes for the effect as they suggest that these biases are not present during acquisition. In other words, in the first encounter with a word or concept, the emotional context around it can have the same effect in the foreign as in the native language. Therefore, the later observed effects might relate to the way in which this information or vocabulary was acquired, rather than with the use of a foreign language itself. In addition, our results have ramifications for our understanding of how we learn a foreign language. In particular, they support the idea that the reduced emotionality found in foreign language contexts has its origins in the way we learn a foreign language —namely, in a classroom and not in our more informal social environment—, rather than being a characteristic of how we process languages that are not our native one [62]. Similar claims have in fact been made by other authors [28]. Although practical applications of this research can be easily extracted, this is not our main goal. Therefore, we only provide the suggestion of looking into manipulating emotionality and semantic context as a way to improve learning, but do not provide practical advice for teachers.

It is worth noting that our study focuses on acquisition for the lexical item in the short term. Although we did not test indicators of word acquisition beyond the item itself and its association with a visual object, we provide evidence on the essential first steps towards word learning. Future research should expand our results to other areas of word acquisition such as semantics and word use. Despite the limitations of the experiments, this study provides ecological validity insofar as participants were taught new words in the same way they might learn them naturally in multilingual classroom contexts: by reading a textbook or asking for the definition of the word. Nevertheless, further research using even more naturalistic situations— e.g., a classroom setting, but maintaining the same level of experimental control—, is necessary in order to understand learning in these situations in more depth.

## Conclusions

The current results get us one step closer to understanding the intricacies of language learning by non-balanced bilinguals and the complex relationship between emotionality and each of the languages they know. Our results point to an increased importance of chunking information— or splitting it up into multiple sessions—in the foreign language with respect to the native language. We highlight the importance of emotional semantic context when learning new words, equally in both the native and foreign language. And, finally, although not our main goal, this provides a practical teaching tool, where simply by adding a positive valence to texts, teachers might be able to help students improve their memory and learning of new vocabulary.

## Supporting information

**S1 Fig. Example trial from norming task.** Each description was given followed by the questions "How positive or negative is the description of this object?" [¿Cómo de positiva o negativa es la descripción de este objeto?] and "How intense is the description of this object?" [¿Cómo de intensa es la descripción de este objeto?] with a clarification underneath saying "Please select how intense the emotional activation you feel is, regardless of whether the description is positive or negative" [Por favor selecciona cómo de intensa es la activación emocional que sientes, más allá de que la descripción sea positiva o negativa]. After each question there was a dropdown menu. In the valence question the answer options were very negative, somewhat negative, neutral, somewhat positive, and very positive [muy negativa, algo negativa, neutral, algo positiva, muy positiva]. For the arousal question, the options were: not at all intense, a little intense, somewhat intense, very intense, extremely intense [para nada intensa, un poco intensa, bastante intensa, muy intensa, extremadamente intensa].
(PDF)

**S1 Table. Pseudowords and their average bigram frequencies.** The following is a list of all the pseudowords used for the main tasks—learning and testing—by type, i.e. whether they were used as learn/test stimuli or as foils. Reported are the average token-type bigram frequencies for the word as reported by B-Pal [40] and N-Watch software [41].
(PDF)

**S1 Appendix. ANOVAs and comparison with LME results.**
(PDF)

## Acknowledgments

We thank Julen Cristti© for the creation of part of the materials that were used in this set of experiments.

## Author Contributions

**Conceptualization:** Candice Frances, Angela de Bruin, Jon Andoni Duñabeitia.

**Data curation:** Candice Frances.

**Formal analysis:** Candice Frances.

**Methodology:** Candice Frances, Angela de Bruin, Jon Andoni Duñabeitia.

**Supervision:** Angela de Bruin, Jon Andoni Duñabeitia.

**Visualization:** Candice Frances.

**Writing – original draft:** Candice Frances.

**Writing – review & editing:** Candice Frances, Angela de Bruin, Jon Andoni Duñabeitia.

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
