## [Decision Letter · Decision Letter 0]

29 May 2020

PONE-D-20-07595

The effects of language and emotionality of stimuli on vocabulary learning

PLOS ONE

Dear Dr. Frances,

Thank you for submitting your manuscript to PLOS ONE. After careful consideration, we feel that it has merit but does not fully meet PLOS ONE’s publication criteria as it currently stands. Therefore, we invite you to submit a revised version of the manuscript that addresses the points raised during the review process.

I am sorry to have kept you waiting. Your manuscript (PONE-D-20-07595) was reviewed by three experts. As you will see below, reviewers commented that the study may have some merits but that the manuscript needs to be revised significantly before accepted for publication. As an academic editor, I myself read the manuscript and believe that all the comments from the reviewers are important. My editorial decision is therefore "Major Revision" and request that you respond to all of their comments. As for the mixed-effects modeling suggested by Reviewer 1, I myself believe that it is a useful approach. Because not all results seem to be solid, the results may change with subjects and items included simultaneously as random effects. I strongly encourage you to report such follow-up analyses (in an appendix) and minimally request that you submit raw trial-level data in a long format. 

We look forward to receiving your revised manuscript.

Kind regards,

Koji Miwa, Ph.D.

Academic Editor

PLOS ONE

Journal Requirements:

2. Please provide additional information about the participant recruitment method and the demographic details of your participants. In particular, please add information about the database from which the participants were recruited and whether your sample can be considered representative of a larger population.

3. We note that Figures 1 and 4 in your submission contain copyrighted images. All PLOS content is published under the Creative Commons Attribution License (CC BY 4.0), which means that the manuscript, images, and Supporting Information files will be freely available online, and any third party is permitted to access, download, copy, distribute, and use these materials in any way, even commercially, with proper attribution. For more information, see our copyright guidelines: http://journals.plos.org/plosone/s/licenses-and-copyright.

1.    You may seek permission from the original copyright holder of Figures 1 and 4 to publish the content specifically under the CC BY 4.0 license.

Additional Editor Comments (if provided):

Reviewers' comments:

Reviewer's Responses to Questions

**Comments to the Author**

1. Is the manuscript technically sound, and do the data support the conclusions?

Reviewer #1: No

Reviewer #2: Partly

Reviewer #3: Partly

2. Has the statistical analysis been performed appropriately and rigorously? 

Reviewer #1: No

Reviewer #2: Yes

Reviewer #3: Yes

3. Have the authors made all data underlying the findings in their manuscript fully available?

Reviewer #1: No

Reviewer #2: Yes

Reviewer #3: Yes

4. Is the manuscript presented in an intelligible fashion and written in standard English?

Reviewer #1: Yes

Reviewer #2: No

Reviewer #3: Yes

5. Review Comments to the Author

Reviewer #1: This paper pairs pictures of novels objects with neutral or positive descriptions and a pseudoword label, then has pariticpants perform tasks involving recall of the pseudoword label. The language of the passages is manipulated between subjects. It finds that positive semantic content in definitions enchances memory for peudowords forms in a new/old and name matching tasks, and 1st or second language status does not influence performance.

I find the paper has some minor scientific merit, but I have sufficient concerns about the statistical treatment to be skeptical of the results. As well, I find the post-tasks to be shallow indicators of vocabulary learning, and this is not discussed in the current manuscript. The third post-task, the characteristic task, yields a null result but is barely acknowledged or discussed. Taken together, I feel the manuscript needs major revisions to have sufficient scientific merit to be published in PLOS.

Statistical Concerns:

The main concern is that the analysis is done on responses averaged to the participant level. No analysis is done at the item level. This falls short of the statistical state of the art in psycholinguistics, which is to perform a mixed effects analsys of raw trial-level responses, with the inclusion of crossed random intercepts of, at least, subjects and items (Baayen, Davidson & Bates, 2008). Neglecting the issue of variation at the level of items commits the "language as fixed effects fallacy" (Clark, 1973) i.e. the analyst is implicitly assuming that the linguistic stimuli in the experiment exhaust their population, which is trivially false in this case (for a broader discussion in psychology see Yarkoni, 2019). Subject-level averaging of responses comes at the expense of statistical power - which is a concern in paper case, given that the authors test for numerous interactions in a very small sample. As well, the stimuli in this experiment are particularly complex: there are seemingly three souces of potential stimulus level random variance: the images, the pseudowords, and the passages. Therefore, an ideal statistical treatment would have started with random intercepts for all these factors (provided the passages, pseudowords, and images were randomly intermixed). It is worth noting that that while this may seem a significiant increase in model complexity, it is not --- amounting to at most 3 additional expended degrees of freedom, while moving to unaveraged data will provide many more degrees of freedom back. Likewise, an analysis on participant level averaged responses does not consider whether the effect of context may vary between subjects. A mixed-effects analysis incorporating a random slope of condition would allow for explicit modelling of this possibility, and achieves a better type I error rate (Barr et al, 2013; Matuschek et al, 2017). Similarly, it is worth noting that a properly specified mixed effect model is also more conservative than subject level averages, despite it's additional complexity. Lacking a full mixed-model analysis, I feel skeptical the results would generalize to other passages, objects, or psuedowords, whether the small observed effects would remain statistically significiant, and whether the interactions would remain null. Checking the OSF repository given by the authors, it appears no trial level data is present, precluding others from doing a state of the art analysis of their own.

As well, I'm curious as to why language was manipulated as between subjects in first experiment. This seems to pass up the opportunity to use repeated measurements to allow participants to be their own controls. Considering the authors went to much work to match the participants between groups, as well as gather individual differences measures, choosing not to do a mixed model strikes me as quite a missed opportunity, since it would allow language to be a within-subjects factor, do away with the need for matching participants between groups, and allow for any relation between individual difference measures and condition to be tested. While they do this manipulation in study 2, the choice for experiment one is never justified.

What was the distribution of accuracy? Was it normally distributed? Did the authors do any transformations on accuracy in the post tasks?

How big are the observed effect sizes and how much should we care about them? Are they practically significant in a way a language educator would care about?

Conceptual Concerns

As is stands, the post tasks administered to the participants are very shallow indicators of word learning. Word learning involves a complex, ongoing acquisition of links between orthographic and phonological forms and meaning, and meaning is acquired to diffent depths (Perfetti, 2002). Both cued recall and old/new recognition are shallow, more indicative of learning of forms than semantics. A useful post-task would have been a meaning generation task: i.e., to have participants attempt to give the meaning assigned to the object in the passage, given only its name or its picture. The shallowness of these tasks to me also hurts the scientific merit of the conclusions, since knowing if positive context learning of form and semantics would be more valuable. Likewise, it would be interesting to know how positive context may influence production versus comprehension, the current tasks comment only on comprehension.

While the authors use the term emotionality in their paper, their manipulation really has more to do with positive valence, due to the lack of a negative condition. As well, although they note emotion captures attention, it is worth noting that, for negative words, additional attention may come at a cost (negative words capture attention and we devote cognitive resources to potential fight or flight resposnes). Rather than the vague "emotionality" the authors could clarify whether they mean: high arousal, positive valence, or positive and negative valence.

The authors claim their stimuli are naturalistic, but it is worth noting that this experiment is not a naturalistic word learning experiment. It is an explicit word learning study, where participants are instructed to learn the words. This is unlike a natural, incidental word learning environment (such an encountering a novel word during passage reading where word learning is not an explicit goal). Rather than call the experimental or stimuli naturalistic, it would be much more correct to say that it attempts to mimic the context of a classroom. Some sections of the manuscript acknowledge this, but others stray into problematic "naturalistic" claims (e.g. line 120).

The fact that the third task, the characteristic matching task, was null, goes unremarked and undiscussed in the general discussion. This for me is the most interesting task, because it is the most semantic. This task should actually tap into a different, slightly deeper but still shallow, aspect of word learning. Yet, the authors say it is unimportant and is only done as an "attention check." I find this reporting and framing of that post-task selective and problematic. At minimum it needs to be discussed more thoroughly, since it implies that positive context is only enhancing shallow, form-based aspects of word learning, and less so semantic ones. As well, this is the post task where I would expect 1st or 2nd language to potentially matter most. This task is not revisited at all in the general discussion and it should be.

Baayen, R. H., Davidson, D. J., & Bates, D. M. (2008). Mixed-effects modeling with crossed random effects for subjects and items. Journal of memory and language, 59(4), 390-412.

Barr, Dale J., et al. "Random effects structure for confirmatory hypothesis testing: Keep it maximal." Journal of memory and language 68.3 (2013): 255-278.

Clark, H. H. (1973). The language-as-fixed-effect fallacy: A critique of language statistics in psychological research. Journal of verbal learning and verbal behavior, 12(4), 335-359.

Matuschek, H., Kliegl, R., Vasishth, S., Baayen, H., & Bates, D. (2017). Balancing Type I error and power in linear mixed models. Journal of Memory and Language, 94, 305-315.

Perfetti, C. A., & Hart, L. (2002). The lexical quality hypothesis. Precursors of functional literacy, 11, 67-86.

Yarkoni, T. (2019). The generalizability crisis. Psyarxiv.com

Reviewer #2: Overall interesting study that deals with the potentiality of learning effects in a foreign language using positive versus neutral semantic contexts. I believe that the authors have created a novel study that is relevant to the broader conversation of how emotions affect second language acquisition (SLA). This is an area within the field of SLA that, in my opinion, is lacking and merits further inquiry. That said, I have a few additional recommendations for the authors to consider before the paper is accepted for publication:

Some researches have found that valenced words (positive and negative) are processed faster than neutral words in lexical decision tasks (see Ponari et al., 2015; Vinson et al., 2014). The authors even showcase the effectiveness of positive and negative valenced words from a study by Ferré et al., 2015. As such, why did the authors not also include the stimuli with negative descriptions to test for the effects of negative emotionality? Could negatively valenced words/descriptions also provide a similar processing benefit (as with positively valenced words/descriptions) and possibly facilitate learning by participants?

Lines 57-59: Run-on sentence; consider breaking up into two sentences.

Lines 82-83: This statement is confusing; consider revising.

Line 91: This statement is not clear: “Frances et al. [15] took a similar approach to the current one, where participants…” Is “current one” referring to the present study? If so, it should be clarified.

Line 102: What is meant by “…expanding vocabulary in that language.”? What are you referring to here?

Line 111: Do you have a citation for the statement, “…which is analogous to how we actually learn emotional words.”? This claim is unexplained.

Lines 204-209: Did the researchers control for specific individual differences variables, such as foreign language anxiety, motivation, working memory, etc.? If not, how might these factors have influenced the results of the study?

Lines 210-213: What was the average length of exposure during the learning phase of each experiment? Did some participants spend longer amounts of time than others studying the objects and their descriptions? If so, what is the likely effect on learning?

Lines 224/417: Why did the researchers choose to arrange the testing items of the two experiments in these particular orders? Could the presence of the first two tests in Experiment 1, for example, have primed participants for the main task (test 3)? In this case, would task randomization have been better suited?

Line 361: Are the overall results from Experiment 1 tenuous, at best, considering the low sample size for each group (N = 20 and N = 23 for the English and Spanish groups, respectively)? This fact could be presented as a limitation or suggestion for improvement during future research.

Line 525: What do the authors mean by, “…taught in a non-official language.”?

Line 568: What do the authors mean by, “…learning new words and concepts may be more costly in a foreign language…”?

Line 597: A variety of concepts and vocabulary presented in the L2 classroom can only be described using negative descriptions (e.g., illness, death, fear, etc.). Based upon the authors’ conclusions in lines 596-598, how can teachers best present these lexical items in the classroom?

Reviewer #3: See attached document.

This study presents two experiments that examine the effect of language (L1 vs. L2) and emotionality (positive vs. neutral) on the acquisition of new concepts by means of definitions. The two experiments differ in the number of target concepts being studied as well as the presentation of the words (in one or two sessions) and the characteristics of the tasks (e.g., number of options in matching task). Employing two-way mixed-factors ANOVAS, the authors concluded a significant effect of language when the set of target words to be learnt in one go is larger, and a stable significant effect of emotionality, with more positive words being learnt.

The study is interesting, and the design is well thought through. The authors have collected a commendable amount of data and performed some sophisticated analyses. However, the manuscript in its current form would require some changes before it can warrant publication in PLOS ONE. The most prominent issue with the manuscript is that the general focus of the study is unclear. The authors state in the abstract and the discussion that the examination of emotionality can inform the learning of foreign languages, in particular regarding vocabulary. However, there is very little discussion about what pedagogical applications or implications the results might have for vocabulary teaching and learning. Similarly, the review of literature is constrained to only studies on emotionality, but it does not include any review of CLIL, word learning, semantics or how the results from emotionality research can be applied in foreign language learning and teaching. The authors should provide a clearer storyline for the study before it is ready for publication.

These and other issues are described in more detail in the attached document.

6. PLOS authors have the option to publish the peer review history of their article (what does this mean?). If published, this will include your full peer review and any attached files.

Reviewer #1: No

Reviewer #2: No

Reviewer #3: No

---

## [Author Response · Author response to Decision Letter 0]

5 Aug 2020

Dear reviewers and editor, 

First and foremost, we would like to thank the editor and the reviewers for taking the time to assess our paper and for the very insightful comments and suggestions they have provided. The most significant modifications made to the manuscript relate to further clarifying our aims and definitions, justifying our methodological choices, adding the by item analyses (and the LMEs as a supplementary material), clarifying some limitations of our study, and expanding on its interpretation. The literature covered in the introduction has been expanded and clarifications have been added throughout. 

We have addressed each of the reviewers' concerns, one-by-one in the attached file labelled "Response to Reviewers". 

We could like to thank you again for your time and consideration. 

Best regards, 

Candice Frances

Angela de Bruin

Jon Andoni Duñabeitia

---

## [Decision Letter · Decision Letter 1]

10 Sep 2020

PONE-D-20-07595R1

The effects of language and emotionality of stimuli on vocabulary learning

PLOS ONE

Dear Dr. Frances,

Thank you for submitting your manuscript to PLOS ONE. After careful consideration, we feel that it has merit but does not fully meet PLOS ONE’s publication criteria as it currently stands. Therefore, we invite you to submit a revised version of the manuscript that addresses the points raised during the review process.

Thank you for your revision. Your manuscript was evaluated by the same reviewers from the previous round. All three reviewers commented that the manuscript improved, and two reviewers (Reviewers 2 and 3) suggested "Accept." Reviewer 1 suggested "Major Revision" proposing that the mixed-effects model should be included in the main text. As an editor, I do not force you to do so at this stage. However, I do believe that similarities and differences between the ANOVA results and the LMM results should be presented to readers more clearly. I request that such information be included either in the general discussion or in the appendix. 

We look forward to receiving your revised manuscript.

Kind regards,

Koji Miwa, Ph.D.

Academic Editor

PLOS ONE

Reviewers' comments:

Reviewer's Responses to Questions

**Comments to the Author**

1. If the authors have adequately addressed your comments raised in a previous round of review and you feel that this manuscript is now acceptable for publication, you may indicate that here to bypass the “Comments to the Author” section, enter your conflict of interest statement in the “Confidential to Editor” section, and submit your "Accept" recommendation.

Reviewer #1: (No Response)

Reviewer #2: All comments have been addressed

Reviewer #3: All comments have been addressed

2. Is the manuscript technically sound, and do the data support the conclusions?

Reviewer #1: No

Reviewer #2: Yes

Reviewer #3: Yes

3. Has the statistical analysis been performed appropriately and rigorously? 

Reviewer #1: No

Reviewer #2: Yes

Reviewer #3: Yes

4. Have the authors made all data underlying the findings in their manuscript fully available?

Reviewer #1: Yes

Reviewer #2: Yes

Reviewer #3: Yes

5. Is the manuscript presented in an intelligible fashion and written in standard English?

Reviewer #1: Yes

Reviewer #2: Yes

Reviewer #3: Yes

6. Review Comments to the Author

Reviewer #1: I reviewed a previous version of this manuscript. I find the manuscript better framed than the previous version. In my previous review, I suggested two things: 1) to change the analysis to a linear-mixed effects one, and 2) to better justify numerous choices, particularly the exclusion of the third "attention check" task. Unfortunately, the fears motivating me to prefer the mixed effects analysis appear to have been realized. By-item ANOVAs yield inconsistent effects, and though the mixed model analysis has been done, it is put in the supplementary materials, even though it gives I think much better results.

I think the mixed model analysis should be the main one. It is considerably more concise, and actually much more consistent than the by-participant and by-item ANOVAs. As well, I am very distressed that the qualitative differences in effects between these analysis are not remarked on. The reader is merely directed to the supplementary materials, without noting the results are not quite the same. I have created below a table summarizing the results of this analysis.

Experiment Task Dep Var Emotion E Direction Language E Direction E/L Interaction

1 Matching Accuracy Y Pos > Neutral Y Spanish > English N

1 Matching RT N N N

1 Old/New Accuracy N N N

1 Old/New RT Y Pos > Neutral Y Spanish > English N

1 Recall Accuracy N Y Spanish > English N

1 Recall Levenshtein Distance N Y Spanish > English N

2 Matching Accuracy Y Pos > Neutral N N

2 Matching RT Y Pos > Neutral Y Spanish > English N

2 Recall Accuracy N N N

2 Recall Levenshtein Distance N Y Spanish > English N

What is worth noting is that there are differences between tasks and experiments, though the overall qualitative trend when significant are identical, which is not the case in the by-item and by-subject ANOVAs. Looking at the mixed model analysis, I am somewhat encouraged, and think the paper makes a good contribution, but it one we should certainly view the findings with many grains of salt. The effects jump around depending on the dependent variable, experiment, and task. The mixed model analysis is much clearer, however.

With regards to the "attention check" task, I find the justifications for not analyzing and including the data very poor. If the stimuli are not perfectly matched, then at the very least language can serve as a statistical control, to see if the positivity advantage remains. If there are large differences in the difficulty of the items, this might be revealed by inspecting the residual variance of a random effect of item, and seeing if the random adjustments for items are imbalanced across conditions.

Put together, I think the manuscript needs to be revised to make the mixed-model analysis the primary analysis, and the results need to be presented more tentatively. I think making the ANOVA the main analysis actually weakens the authors finding. As well, I would like to see a mixed-model analysis of the "attention check task", with appropriate hedging, rather than merely brushing this task aside.

Reviewer #2: The authors have addressed all of my comments/concerns. My evaluation is to accept the manuscript in its current state.

Reviewer #3: The review of the current manuscript shows that the authors have dedicated significant time and effort to answering and addressing all the issues/comments raised by the reviewers. I would like to thank the authors for these explanations to our questions and for their patience adapting the manuscript accordingly.

As a reviewer, I am satisfied with the authors’ responses to my questions and comments. The authors have successfully addressed the issues/suggestions I identified in the previous round of review, either by adding extra information to the manuscript or by explaining why changes were not appropriate/necessary to be made in the manuscript.

In particular, the authors have now clarified the main aim of the study. It is clear that the paper has a more theoretical focus, although the results can also inform pedagogy. The terminology employed (i.e., emotionality, native vs. foreign language) has also been explicitly described and justified, helping the readability of the paper. Finally, their added discussion on the learning gains and implications for vocabulary acquisition, although brief, is appreciated.

Below are some final, minor suggestions that I would like to make before the paper is published:

“One of the priorities in the current study is to use naturalistic classroom-type stimuli.”

I appreciate that the authors have added classroom-type in response to a comment in the previous round of review regarding how natural stimuli and tasks are. However, I would recommend to further paraphrase this statement slightly, for example: “… to use more naturalistic classroom-type stimuli than in most previous research”.

“We hypothesized that new lexical items would be more difficult to learn in a foreign language—even though the words were exactly the same between languages—…"

When I read this sentence in the second revision I had to stop and read it again, and I think future readers might experience the same situation. The confusing part was “.. the words were exactly the same between languages”. Maybe this can be paraphrased to say “even though the target words in the stimuli were exactly the same between conditions”.

“It is worth nothing that our study focuses on….”

I think that you meant “noticing” in this statement at the end of the Discussion section. One of the most unfortunate tricks played by autocorrectors that you want being published in a manuscript!

Overall, the authors have done a good job with the reviews. I hope to see the paper published soon.

7. PLOS authors have the option to publish the peer review history of their article (what does this mean?). If published, this will include your full peer review and any attached files.

Reviewer #1: No

Reviewer #2: No

Reviewer #3: No

---

## [Author Response · Author response to Decision Letter 1]

17 Sep 2020

First and foremost, we would like to thank the editor and the reviewers once more for taking the time to assess our paper and for the interesting points they raised. The most significant modifications made to the manuscript are including the linear mixed effect models in the main text and adding a comparison between the linear mixed effect models and analysis of variance results in the appendix. In the following pages, we have addressed each of the reviewers’ concerns, one-by-one.

Reviewer 1

I reviewed a previous version of this manuscript. I find the manuscript better framed than the previous version. In my previous review, I suggested two things: 1) to change the analysis to a linear-mixed effects one, and 2) to better justify numerous choices, particularly the exclusion of the third "attention check" task. Unfortunately, the fears motivating me to prefer the mixed effects analysis appear to have been realized. By-item ANOVAs yield inconsistent effects, and though the mixed model analysis has been done, it is put in the supplementary materials, even though it gives I think much better results.

I think the mixed model analysis should be the main one. It is considerably more concise, and actually much more consistent than the by-participant and by-item ANOVAs. As well, I am very distressed that the qualitative differences in effects between these analysis are not remarked on. The reader is merely directed to the supplementary materials, without noting the results are not quite the same. I have created below a table summarizing the results of this analysis.

What is worth noting is that there are differences between tasks and experiments, though the overall qualitative trend when significant are identical, which is not the case in the by-item and by-subject ANOVAs. Looking at the mixed model analysis, I am somewhat encouraged, and think the paper makes a good contribution, but it one we should certainly view the findings with many grains of salt. The effects jump around depending on the dependent variable, experiment, and task. The mixed model analysis is much clearer, however.

We thank the reviewer for these suggestions. In response to this point, we have now changed the manuscript to present the linear mixed effect models as the main analyses and we have provided a comparison between analyses in the supplementary materials (along with the ANOVA results). 

With regards to the "attention check" task, I find the justifications for not analyzing and including the data very poor. If the stimuli are not perfectly matched, then at the very least language can serve as a statistical control, to see if the positivity advantage remains. If there are large differences in the difficulty of the items, this might be revealed by inspecting the residual variance of a random effect of item, and seeing if the random adjustments for items are imbalanced across conditions.

We thank the reviewer for this comment. Nevertheless, we maintain that this task was not built to be analyzed in more detail and would require a level of discussion that would detract from the main results. We fear that presenting these results would provide an inaccurate picture of any possible effects, as the task varies between experiments and the items varied widely in imageability, length, and importance in the text. This task was conceived and carried out as a rough measure of attention to detract from only focusing on the vocabulary and not as a way to assess learning. Furthermore, in Experiment 1 this task does not involve the learned vocabulary at all, meaning that it completely misses the central theme of the paper, namely, vocabulary learning. For these reasons, we believe that including this analysis would weaken the paper rather than provide further insight. 

 

Reviewer #3

Below are some final, minor suggestions that I would like to make before the paper is published:

“One of the priorities in the current study is to use naturalistic classroom-type stimuli.”

I appreciate that the authors have added classroom-type in response to a comment in the previous round of review regarding how natural stimuli and tasks are. However, I would recommend to further paraphrase this statement slightly, for example: “… to use more naturalistic classroom-type stimuli than in most previous research”.

We thank the reviewer for this comment and have now clarified this statement further. 

“We hypothesized that new lexical items would be more difficult to learn in a foreign language—even though the words were exactly the same between languages—…"

When I read this sentence in the second revision I had to stop and read it again, and I think future readers might experience the same situation. The confusing part was “.. the words were exactly the same between languages”. Maybe this can be paraphrased to say “even though the target words in the stimuli were exactly the same between conditions”.

We thank the reviewer for this comment and have now clarified this statement.

“It is worth nothing that our study focuses on….”

I think that you meant “noticing” in this statement at the end of the Discussion section. One of the most unfortunate tricks played by autocorrectors that you want being published in a manuscript!

We thank the reviewer for this comment and have fixed this typo.

---

## [Decision Letter · Decision Letter 2]

23 Sep 2020

The effects of language and emotionality of stimuli on vocabulary learning

PONE-D-20-07595R2

Dear Dr. Frances,

We’re pleased to inform you that your manuscript has been judged scientifically suitable for publication and will be formally accepted for publication once it meets all outstanding technical requirements.

Kind regards,

Koji Miwa, Ph.D.

Academic Editor

PLOS ONE

Additional Editor Comments (optional):

Reviewers' comments:

Reviewer's Responses to Questions

**Comments to the Author**

1. If the authors have adequately addressed your comments raised in a previous round of review and you feel that this manuscript is now acceptable for publication, you may indicate that here to bypass the “Comments to the Author” section, enter your conflict of interest statement in the “Confidential to Editor” section, and submit your "Accept" recommendation.

Reviewer #1: All comments have been addressed

2. Is the manuscript technically sound, and do the data support the conclusions?

Reviewer #1: Yes

3. Has the statistical analysis been performed appropriately and rigorously? 

Reviewer #1: Yes

4. Have the authors made all data underlying the findings in their manuscript fully available?

Reviewer #1: Yes

5. Is the manuscript presented in an intelligible fashion and written in standard English?

Reviewer #1: Yes

6. Review Comments to the Author

Reviewer #1: This revision changes the mixed-model analysis to be the main one in the text. This addresses my main concern from the last revision.

Although I still find the authors justifications for not including data from the attention check task somewhat problematic, I do not feel it is an sufficient issue to prevent publication of the results from the other tasks.

7. PLOS authors have the option to publish the peer review history of their article (what does this mean?). If published, this will include your full peer review and any attached files.

Reviewer #1: No

---

## [Editor Report · Acceptance letter]

28 Sep 2020

PONE-D-20-07595R2 

The effects of language and emotionality of stimuli on vocabulary learning 

Dear Dr. Frances:

I'm pleased to inform you that your manuscript has been deemed suitable for publication in PLOS ONE. Congratulations! Your manuscript is now with our production department. 

Kind regards, 

on behalf of

Dr. Koji Miwa 

Academic Editor

PLOS ONE